# Bipartite chromatin recognition by Hop1 from two diverged Holozoa

Alyssa A Rodriguez[1], Alessandro E Cirulli[1,2], Katie Chau[1], Justin Nguyen[1], Qiaozhen Ye[1], Kevin D Corbett[1,3]

In meiosis, ploidy reduction is driven by a complex series of DNA breakage and recombination events between homologous chromosomes, orchestrated by meiotic HORMA domain proteins (HORMADs). Meiotic HORMADs possess a central chromatin binding region (CBR) whose architecture varies across eukaryotic groups. Here, we determine high-resolution crystal structures of the meiotic HORMAD CBR from two diverged aquatic Holozoa, *Schistosoma mansoni* and *Patiria miniata*, which reveal tightly associated plant homeodomain (PHD) and winged helix-turn-helix (wHTH) domains. We show that PHD–wHTH CBRs bind duplex DNA through their wHTH domains, and identify key residues that disrupt this interaction. Combining experimental and predicted structures, we show that the CBRs' PHDs likely interact with the tail of histone H3, and may discriminate between unmethylated and trimethylated H3 lysine 4. Finally, we show that Holozoa Hop1 CBRs bind nucleosomes in vitro in a bipartite manner involving both the PHD and wHTH domain. Our data reveal how meiotic HORMADs with PHD–wHTH CBRs can bind chromatin and potentially discriminate between chromatin states to drive meiotic recombination to specific chromosomal regions.

## Introduction

Meiosis is a specialized two-stage cell division program that decreases chromosome number (ploidy) by half to produce haploid gametes or spores (Zickler & Kleckner, 2023). Ploidy reduction in meiosis requires the introduction of programmed DNA double-strand breaks (DSBs) along each chromosome, followed by the repair of these breaks as interhomolog crossovers or chiasmata. Crossovers enable the proper association and then segregation of homologous chromosomes from one another in the meiosis I division, and also drive genetic diversity in offspring (Raghavan & Hochwagen, 2025). During early prophase of meiosis I, each pair of replicated sister chromosomes condenses into a linear array of chromatin loops around a protein assembly termed the meiotic chromosome axis (Ur & Corbett, 2021; Ito & Shinohara, 2022). The chromosome axis is highly conserved across eukaryotes and typically contains three major components: HORMA domain–containing proteins (HORMADs) (Gu et al, 2022; Prince & Martinez-Perez, 2022), axis core proteins that form filamentous assemblies through coiled-coil domains (West et al, 2019), and DNA binding cohesin complexes with at least one meiosis-specific subunit (Ur & Corbett, 2021; Sakuno & Hiraoka, 2022). HORMADs play multiple roles in meiotic prophase, including recruiting DSB machinery to form DNA breaks at specific locations called DSB hotspots (Woltering et al, 2000; Panizza et al, 2011; Milano et al, 2024). Axis core proteins recruit HORMADs to chromatin through conserved "closure motifs" and cooperate with cohesin complexes to assemble regular chromatin loop arrays, supporting recombination and homolog pairing/synapsis (Woltering et al, 2000; Kim et al, 2014; West et al, 2019; Ur & Corbett, 2021; Gu et al, 2022).

The *Saccharomyces cerevisiae* meiotic HORMAD protein Hop1 was recently found to contain a central chromatin binding region (CBR) that specifically binds nucleosomes, the fundamental unit of eukaryotic chromatin (Heldrich et al, 2022; Milano et al, 2024). The *S. cerevisiae* Hop1 CBR comprises a PHD (plant homeodomain) tightly packed against a variant winged helix-turn-helix domain (wHTH), plus an extended C-terminal region (HTH-C) that drapes across both domains. PHDs typically bind histone tails, often histone H3, and can specifically recognize modifications like trimethylation at residue lysine 4 (H3K4me3) (Sanchez & Zhou, 2011). Notably, the canonical lysine binding pocket in the PHD of *S. cerevisiae* and closely related budding yeast Hop1 CBRs is poorly conserved, suggesting that these proteins do not bind histone tails (Milano et al, 2024). Indeed, a structure of the *S. cerevisiae* Hop1 CBR bound to a nucleosome shows that Hop1 binds the outer surface of the highly bent nucleosomal DNA through a non-canonical DNA binding surface involving both the PHD and wHTH domain (Milano et al, 2024). The *S. cerevisiae* Hop1 CBR mediates a general enrichment of chromosome axis proteins in nucleosome-rich regions of the genome (Milano et al, 2024), but apparently does not recognize a specific chromatin state.

[1]Department of Cellular and Molecular Medicine, University of California San Diego, La Jolla, CA, USA [2]Department of Chemistry and Biochemistry, University of California San Diego, La Jolla, CA, USA [3]Department of Molecular Biology, University of California San Diego, La Jolla, CA, USA

Correspondence: kcorbett@ucsd.edu
Alyssa A Rodriguez's present address is Occidental College, Los Angeles, CA, USA

Although *S. cerevisiae* and other budding yeast possess a Hop1 CBR with a PHD–wHTH–HTH-C architecture, HORMAD proteins in other eukaryotes possess CBRs with distinct architectures (Milano et al, 2024). Although HORMADs from major animal model organisms like *M. musculus* and *C. elegans* lack the CBR entirely, HORMADs in some Holozoa possess a PHD–wHTH CBR (lacking the HTH-C extension found in budding yeast). Notably, sequence alignments of PHD–wHTH CBRs from Holozoa show that these proteins' PHD lysine binding pockets are highly conserved, suggesting that they may bind chromatin in a manner distinct from *S. cerevisiae* Hop1, and potentially recognize particular histone modifications (Milano et al, 2024). Meanwhile, HORMAD proteins in Archaeplastida (plants and algae) often possess a CBR with tandem wHTH domains (Milano et al, 2024). These findings suggest that HORMAD CBRs from different eukaryotic groups recognize chromatin through distinct mechanisms.

Here, we show by x-ray crystallography that the Hop1 CBRs from two diverged Holozoa—the blood fluke *Schistosoma mansoni* and the sea star *Patiria miniata*—are structurally similar to one another, comprising tightly packed PHD and wHTH domain. In contrast to the budding yeast Hop1 CBR, we find that *S. mansoni* and *P. miniata* Hop1 CBRs bind DNA via their wHTH domains through a canonical DNA binding interface. We further find that the *P. miniata* Hop1 CBR binds nucleosomes and that both the PHD and wHTH domain contribute to binding, suggesting a bipartite recognition mechanism for chromatin binding. Overall, our data show that despite the architectural variability of meiotic HORMAD CBRs, they broadly share the ability to recognize and bind chromatin.

# Results and Discussion

### Structure of PHD–wHTH Hop1 CBRs from two diverged Holozoa

Our prior study of the budding yeast Hop1 CBR showed that this module, which comprises tightly packed PHD, wHTH, and HTH-C domains, binds bent nucleosomal DNA in a noncanonical manner through a composite interface spanning its PHD and wHTH domain (Milano et al, 2024). Our phylogenetic analysis also revealed that many Holozoa possess Hop1 CBRs with PHD and wHTH domain, and lack the HTH-C extension (Fig 1A and B) (Milano et al, 2024). To determine these proteins' structures and mechanisms of chromatin binding, we expressed and purified the Hop1 CBRs of two diverged aquatic Holozoa: *Schistosoma mansoni* (blood fluke) and *P. miniata* (star fish). We crystallized and determined high-resolution x-ray crystal structures of both proteins using single-wavelength anomalous diffraction (SAD) methods with the anomalous diffraction of endogenously bound $Zn^{2+}$ ions. The structures show CBRs with PHD and wHTH domain tightly packed on one another in an overall configuration similar to that of the budding yeast Hop1 CBR, but lacking the HTH-C extension seen in that protein (Figs 1C–F and S1A–D). The *S. mansoni* and *P. miniata* Hop1 CBRs are 43% identical at the amino acid level, and are nearly identical in structure, with an overall Cα root mean squared displacement (r.m.s.d.) of 1.4 Å over 136 residues. Within the CBR, the N-terminal PHD shows two highly conserved zinc coordination

sites, one with three cysteines and one histidine (*S. mansoni* Hop1 residues C284, C286, H306, and C309) and the second with four cysteines (*S. mansoni* Hop1 C298, C301, C324, and C327) (Fig 1D and F).

### The Hop1 CBR wHTH domain binds DNA

wHTH domains canonically bind DNA via insertion of an α-helix (recognition helix 3) into the major groove of DNA, plus interaction of the "wing" motif with the neighboring minor groove (Lai et al, 1993; Gajiwala & Burley, 2000). Structure-similarity searches using DALI (Holm, 2022) and Foldseek (van Kempen et al, 2024) revealed that the *S. mansoni* and *P. miniata* Hop1 CBR wHTH domains are more structurally similar to canonical DNA binding wHTH domains than to the budding yeast Hop1 CBR, which binds DNA in a non-canonical manner (Milano et al, 2024). We modeled DNA-bound *S. mansoni* and *P. miniata* Hop1 CBRs by overlaying their wHTH domains with a structure of DNA-bound *H. sapiens* hepatocyte nuclear factor-3 (HNF-3; PDB ID 1VTN) (Clark et al, 1993). The resulting models (Fig 2A and B) show that the Hop1 CBRs can accommodate DNA on a positively charged surface (Fig 2C and D), supporting the idea that these proteins bind DNA via the canonical wHTH-DNA binding interface. To test for DNA binding, we used a fluorescence polarization–based assay with a fluorescein-labeled double-stranded DNA oligonucleotide and His-GST–tagged Hop1 CBRs. We detected DNA binding by both *S. mansoni* and *P. miniata* Hop1 CBRs ($Kd$ = 16 µM for both; Fig 2E and F). Mutation of arginine residues in each protein's predicted DNA binding surface (R389 and R407 for *S. mansoni*; R463, R477, and R463 for *P. miniata*) to alanine resulted in a complete loss of DNA binding (Fig 2E and F). Overall, these data show that in contrast to budding yeast Hop1 CBRs, *S. mansoni* and *P. miniata* Hop1 CBRs bind DNA via a canonical wHTH-DNA binding interface (Fig S2A–C).

### The Hop1 CBR PHD likely binds a histone tail

PHDs have diverse roles in chromatin remodeling and transcription, and typically function by binding a modified or unmodified lysine residue on a histone tail in a conserved hydrophobic pocket. PHDs predominantly bind trimethylated histone H3 lysine 4 (H4K4me3), though some PHDs bind unmethylated (me0), monomethylated (me1), or dimethylated (me2) H3K4 or other histone tail residues like H3 arginine 2 or lysine 14 (Sanchez & Zhou, 2011). The lysine binding pocket of PHDs typically comprises an "aromatic cage" with hydrophobic or aromatic residues in four conserved positions termed I, II, III, and IV (Ramón-Maiques et al, 2007; Sanchez & Zhou, 2011). Position I is typically a tryptophan, and position II is often a methionine, which is proposed to contribute to binding through dispersion forces and sulfur–pi interactions (Albanese & Waters, 2021). Positions III and IV are more variable, with position III mostly hydrophobic and position IV either hydrophobic or negatively charged.

Structure-based similarity searches with DALI (Holm, 2022) and Foldseek (van Kempen et al, 2024) revealed that *S. mansoni* and *P. miniata* Hop1 CBR PHDs are most closely related to a set of PHDs that bind H3K4me3 (*S. cerevisiae* Set3 [PDB ID 5TDW] [Gatchalian et al, 2017], *H. sapiens* MLL5 [PDB ID 4L58] [Ali et al, 2013]) or

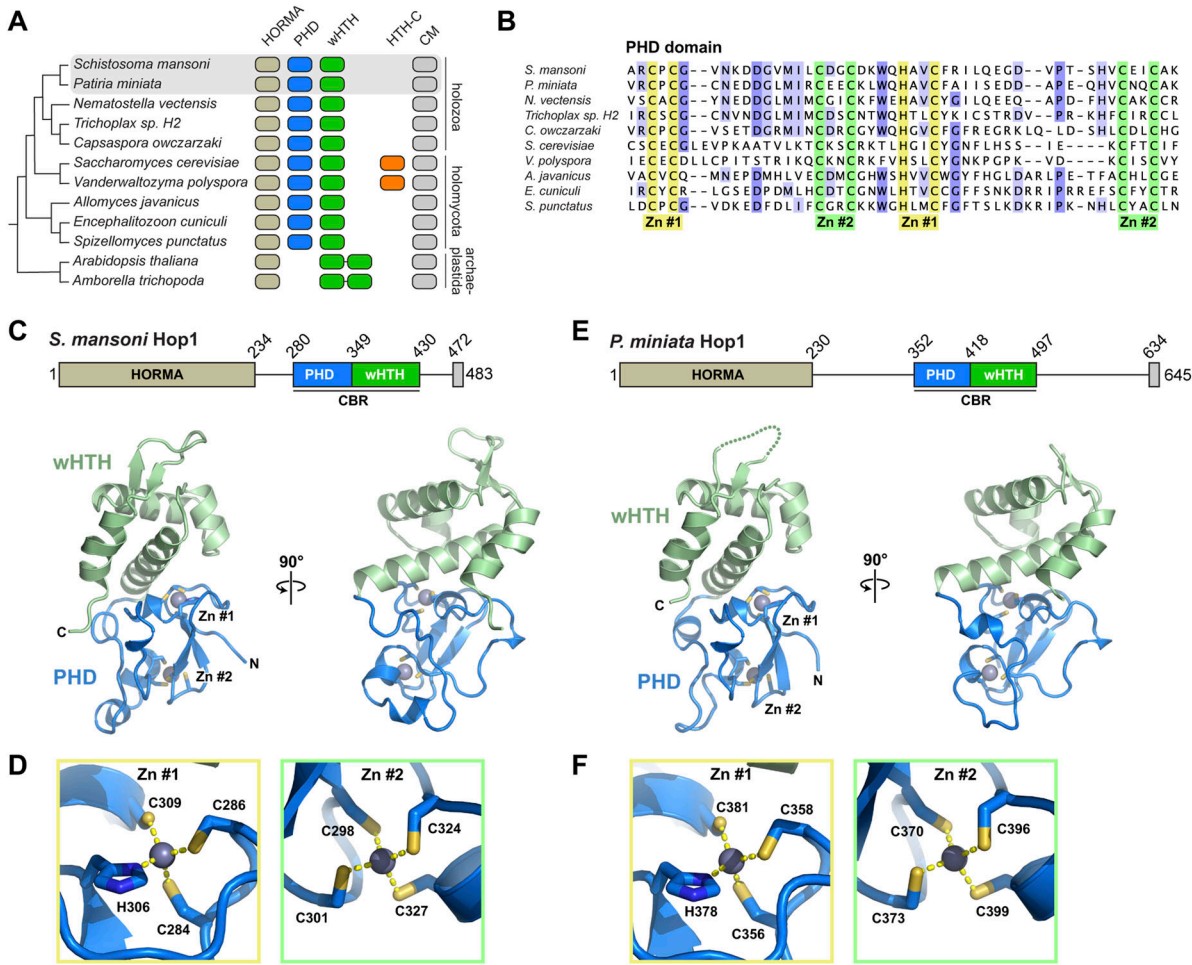

**Figure 1. Structures of the *S. mansoni* and *P. miniata* Hop1 chromatin binding regions (CBRs).**
**(A)** Domain architecture of Hop1 from selected eukaryotes that encode a CBR, with the presence of a HORMA domain indicated in light brown, a plant homeodomain (PHD) in blue, winged helix-turn-helix (wHTH) in green (two organisms possess tandem wHTH domains and lack a PHD), HTH-C C-terminal CBR extension in orange, and a HORMA domain binding closure motif (CM) in gray. The phylogenetic tree (left) is based on overall phylogenetic relationships between these species (Milano et al, 2024). **(A, B)** Sequence alignment of PHDs from organisms shown in (A). Conserved Zn²⁺ binding residues are highlighted in yellow (Zn #1) and green (Zn #2). NCBI accession numbers for displayed sequences are XP_018653011 (*S. mansoni*), XP_038051536.1 (*P. miniata*), XP_001639673 (*N. vectensis*), RDD46648 (*Trichoplax* sp. H2), XP_004363476 (*C. owczarzaki*), NP_012193 (*S. cerevisiae*), XP_001642921 (*V. polyspora*), KAJ3363264 (*A. javanicus*), CAD25118 (*E. cuniculi*), and KND01595 (*S. punctatus*). **(C)** Crystal structure of the *Schistosoma mansoni* Hop1 CBR, with PHD in blue, wHTH domain in green, and bound Zn²⁺ ions in gray spheres. **(D)** Close-up view of bound Zn²⁺ ions for the *Schistosoma mansoni* Hop1 CBR, with Zn²⁺-coordinating residues shown as sticks. **(E)** Crystal structure of the *Patiria miniata* Hop1 CBR, with PHD in blue, wHTH domain in green, and bound Zn²⁺ ions in gray spheres. **(F)** Close-up view of bound Zn²⁺ ions for the *Patiria miniata* Hop1 CBR, with Zn²⁺-coordinating residues shown as sticks.

H3K4me2 (*H. sapiens* PHF20 [PDB ID 5TAB] [Klein et al, 2016], *H. sapiens* ASH1L [PDB IF 7Y0I] [Yu et al, 2022]). Comparison of these proteins' lysine binding pockets with our Hop1 CBR structures revealed strong similarity: position I is a tryptophan in all cases, position II is a methionine in most cases (threonine in Set3), position III is typically a small hydrophobic residue (alanine or valine in Hop1; isoleucine, valine, or threonine in the others), and position IV is negatively charged (aspartate or glutamate; Fig 3A–D).

We next performed a sequence alignment of a broad set of PHDs found in eukaryotic Hop1 CBRs (Milano et al, 2024). This alignment shows high conservation in positions I-IV among Hop1 CBRs from Holozoa and Holomycota, with the exception of the budding yeast family whose CBRs possess the HTH-C extension (Fig 3E). In Hop1 CBRs with the PHD–wHTH architecture, position I is highly

conserved as a tryptophan, position II is a methionine or leucine, position III is a small hydrophobic residue (alanine, valine, or isoleucine), and position IV is most often an aspartate. These data show that with the exception of budding yeast Hop1 CBRs, which are known to bind nucleosomes in a noncanonical manner (Milano et al, 2024), the PHD lysine binding pockets in Hop1 CBRs are highly conserved and similar to PHDs known to bind di- or trimethylated H3K4.

We attempted to directly detect binding between purified *S. mansoni* or *P. miniata* Hop1 CBRs and diverse modified and unmodified histone tail peptides, without success. As an alternative, we used AlphaFold 3 (Abramson et al, 2024) to predict the structures of 20 different Hop1 CBRs with the N-terminal tails of histones H2A, H2B, H3 (unmodified or trimethylated at lysine 4), and

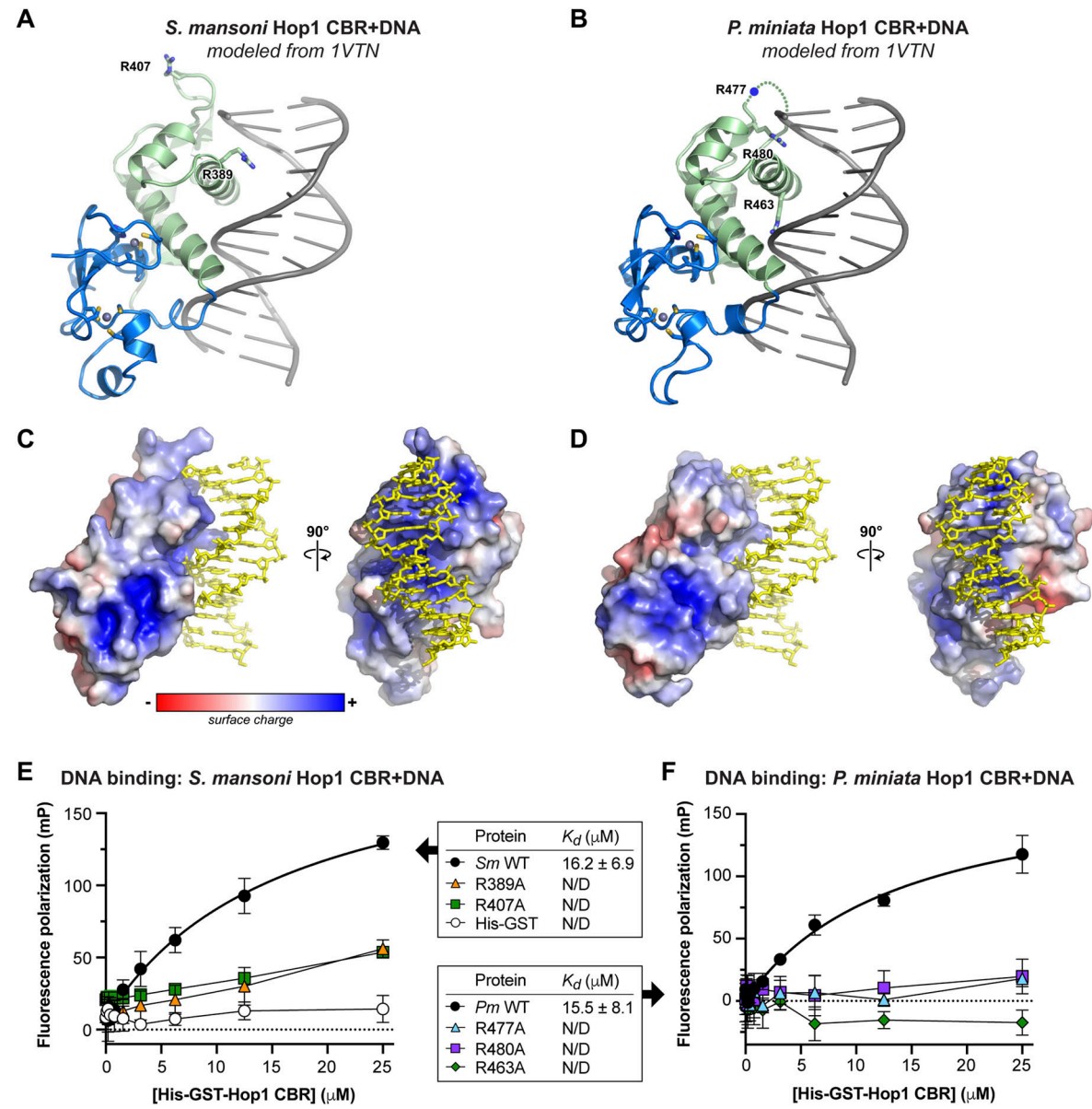

**Figure 2. Holozoa Hop1 chromatin binding regions (CBRs) bind DNA through their wHTH domains.**
**(A)** Crystal structure of *S. mansoni* Hop1 CBR modeled with DNA (shown in gray) by overlaying with a crystal structure of a DNA-bound wHTH domain (HNF-3; PDB ID 1VTN) (Clark et al, 1993). Residues R389 and R407 are shown as sticks. See Fig S2A–C for comparison of DNA binding surfaces between *S. mansoni* Hop1 and *S. cerevisiae* Hop1. **(A, B)** Crystal structure of *P. miniata* Hop1 CBR modeled with DNA as in (A). Residues R463 and R480 are shown as sticks; the location of R477 (in the disordered "wing") is indicated with a blue circle. **(C)** Two views of an electrostatic surface of *S. mansoni* Hop1 CBR (calculated with APBS) (Jurrus et al, 2018), with modeled DNA shown in yellow sticks. **(D)** Two views of an electrostatic surface of *P. miniata* Hop1 CBR (calculated with APBS), with modeled DNA shown in yellow sticks. **(E)** Fluorescence polarization assay showing binding of His-GST–tagged *S. mansoni* Hop1 CBR to a 20-base pair double-stranded DNA. WT Hop1 is indicated in black circles, R389A in orange triangles, R407A in green squares, and His-GST negative control in white circles. Error bars represent the SD from triplicate measurements. **(F)** Fluorescence polarization assay showing binding of His-GST–tagged *P. miniata* Hop-1 CBR to a 20-base pair double-stranded DNA. WT Hop1 is indicated in black circles, R477A in cyan triangles, R480A in purple squares, and R463A in green diamonds. Error bars represent the SD from triplicate measurements.

H4, then systematically analyzed the confidence of the predicted interaction using the AlphaFold ipTM (interface predicted template modeling) score (Figs 3F and S3A). Our test set included 17 Hop1 CBRs with PHD and wHTH domain (including *S. mansoni* and *P. miniata*), and three budding yeast Hop1 CBRs with PHD, and wHTH and HTH-C domains. Because PHD–wHTH–HTH-C CBRs bind nucleosomes in a noncanonical manner (Milano et al, 2024) and do

not possess a conserved lysine binding pocket in their PHDs (Fig 3E), these proteins served as a negative control. Across all 17 Hop1 CBRs with PHD–wHTH architecture, the most confidently predicted histone tail interactions were with histone H3 with tri-methylated lysine 4 (H3K4me3). All 17 predictions showed an ipTM score above the medium-confidence threshold of 0.6, and four scored above the high-confidence threshold of 0.8 (Fig 3F) (Kim

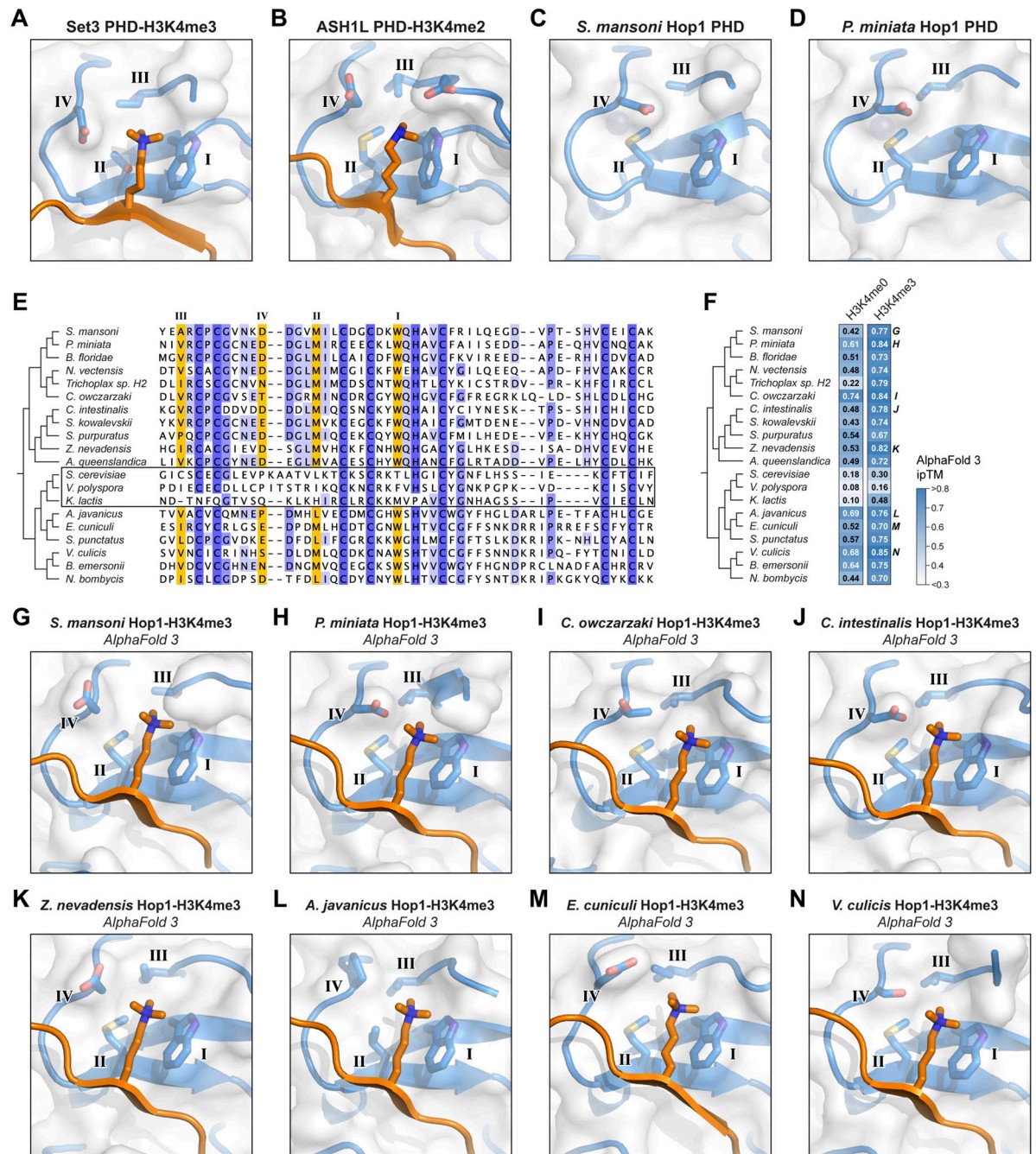

**Figure 3. Hop1 chromatin binding regions (CBRs) likely bind a histone tail.**

**(A)** Close-up of *S. cerevisiae* Set3 plant homeodomain (PHD) (blue) binding to trimethylated lysine 4 of histone H3 (H3K4me3; orange) (PDB ID 5TDW) (Gatchalian et al, 2017), with canonical PHD lysine binding pocket motif residues labeled I, II, III, and IV (Sanchez & Zhou, 2011). **(B)** Close-up of *H. sapiens* ASH1L PHD (blue) binding to H3K4me2 (orange) (PDB ID 7Y0I) (Yu et al, 2022), with canonical PHD lysine binding pocket motif residues labeled I, II, III, and IV. **(C)** Close-up of the *S. mansoni* Hop1 CBR PHD lysine binding pocket with canonical PHD lysine binding pocket motif residues labeled I, II, III, and IV. **(D)** Close-up of the *P. miniata* Hop1 CBR PHD lysine binding pocket with canonical PHD lysine binding pocket motif residues labeled I, II, III, and IV. **(E)** Sequence alignment of 20 Hop1 CBR PHDs, with canonical PHD lysine binding pocket motif residues I, II, III, and IV highlighted in orange. Boxed are three Hop1 CBRs from budding yeast that show the PHD–wHTH–HTH-C architecture, for which the lysine binding pocket is not well conserved (Milano et al, 2024). See the Materials and Methods section for NCBI accession numbers. **(F)** Summary of AlphaFold 3 predictions of the interaction between Hop1 CBRs and the histone H3 N-terminal tail, either unmodified (H3K4me0) or with lysine 4 trimethylated (H3K4me3). Shown are ipTM scores for the most confident of five models generated; see all scores and scores for histones H2A, H2B, and H4 tails in Fig S3A. Models above the medium-confidence ipTM cutoff of 0.6 are indicated with white text. **(G, H, I, J, K, L, M, N)** Close-up of 8 Hop1 CBR PHDs (blue) binding H3K4me3, as predicted by AlphaFold 3. Canonical PHD lysine binding pocket motif residues labeled I, II, III, and IV. See Fig S3B–I for the same views colored by confidence (pLDDT), and Fig S4 for complete models and PAE plots.

et al, 2024 *Preprint*). All 17 predictions showed the modified H3K4me3 residue docked in the CBR's lysine binding pocket, consistent with known structures of PHD–H3K4me3 complexes (Figs 3G–N, S3B–I, and S4). Meanwhile, the three Hop1 CBRs from budding yeast consistently showed low-confidence predictions with all histone tails, including the H3 tail with H3K4me3 (Figs 3F and S3A). Overall, these predictions are consistent with a model in which PHD–wHTH CBRs can bind a histone tail residue, potentially trimethylated H3K4, as part of its chromatin recognition mechanism.

### Hop1 CBRs from Holozoa bind nucleosomes

We previously showed that the budding yeast Hop1 PHD–wHTH–HTH-C CBR binds nucleosomes via a noncanonical surface, recognizing the bent nucleosomal DNA and not contacting the histone proteins themselves (Milano et al, 2024). To test whether the Holozoa Hop1 PHD–wHTH CBR domain binds nucleosomes, we performed electrophoretic mobility shift assays (EMSAs) with recombinant *P. miniata* Hop1 CBR and nucleosome core particles (NCPs) assembled from *Xenopus laevis* histones and DNA containing the Widom 601 nucleosome-positioning sequence, either unmodified or containing a structural analog of the H3K4me3 modification (Simon et al, 2007). We found that the *P. miniata* Hop1 CBR binds nucleosomes with an affinity in the low micromolar range and that alanine mutants in the PHD aromatic cage (position I residue W376) and the wHTH domain DNA binding surface (R480) each show reduced nucleosome binding (Fig 4A and B). WT *P. miniata* Hop1 CBR bound to unmodified and H3K4me3 nucleosomes with roughly equivalent affinity (Fig 4C), indicating the Hop1 CBR alone does not strongly discriminate between H3K4 methylation states in vitro. We also performed EMSAs with the isolated Widom 601 DNA, and observed that the wHTH domain mutant R480A, but not the PHD mutant W376A, reduced DNA binding (Fig 4D). We were unable to perform the same analysis with the *S. mansoni* Hop1 CBR because of poor solubility of the aromatic cage mutant (W304A; not shown). Overall, these data are consistent with bipartite recognition of nucleosomes by the *P. miniata* Hop1 CBR, with both the PHD and wHTH domain contributing to the interaction.

In conclusion, we show that meiotic HORMAD CBRs with the PHD–wHTH architecture recognize chromatin through a bipartite interface, with histone tail binding by the PHD and DNA binding by the wHTH domain. Although our in vitro assays do not show strong discrimination between methylated and unmethylated H3K4 by the Hop1 CBR, our AlphaFold predictions nonetheless suggest that in vivo, this domain may preferentially bind chromatin with the H3K4me3 mark. Given that meiotic HORMADs promote DSB formation by recruiting DSB machinery, HORMAD CBRs with the PHD–wHTH architecture may therefore represent one mechanism for targeting DSBs to H3K4me3 chromatin. Testing this idea in a genetically tractable model organism that possesses a PHD–wHTH CBR, such as the sea star *P. miniata* (Meyer & Hinman, 2022; Zueva & Hinman, 2023 *Preprint*), is therefore an important avenue for future work. Relatedly, whether and how wHTH–wHTH CBRs found in meiotic HORMADS in Archaeplastida recognize particular chromatin states will also be important to test.

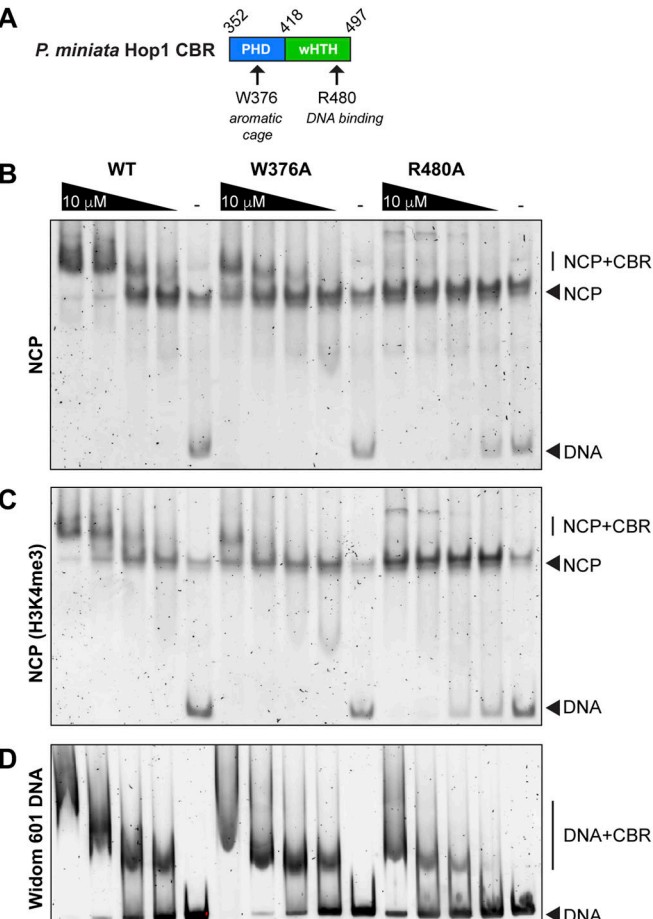

**Figure 4. Bipartite nucleosome recognition by the *P. miniata* Hop1 chromatin binding region (CBR).**
**(A)** Schematic of the *P. miniata* Hop1 CBR and residues mutated in the plant homeodomain aromatic cage (W376) and wHTH domain DNA binding surface (R480). **(B)** Representative electrophoretic mobility shift assay (EMSA) with *P. miniata* Hop1 CBR (WT, W376A, or R480A at 10 μM and twofold dilutions) and 50 nM nucleosome core particles. **(C)** Representative EMSA with *P. miniata* Hop1 CBR and nucleosome core particles containing a H3K4me3 modification analog. **(D)** Representative EMSA with *P. miniata* Hop1 CBR and Widom 601 DNA.

## Materials and Methods

### Cloning, expression, and protein purification

Codon-optimized gene blocks encoding the CBRs of *S. mansoni* Hop1 (NCBI XP_018653011, residues 279–430) and *P. miniata* Hop1 (NCBI XP_038051536.1, residues 352–499) were purchased from GenScript and cloned into the UC Berkeley Macrolab vector 2B-T (#29666; Addgene), which encodes an N-terminal TEV protease–cleavable His$_6$-tag; 2C-T (#29706; Addgene), which encodes an N-terminal TEV protease–cleavable His$_6$-maltose binding protein (MBP) tag; and 2G-T (#29707; Addgene), which encodes an N-terminal TEV protease–cleavage His$_6$-GST tag. Plasmids were transformed into Rosetta 2 (DE3) pLysS *E. coli*–competent cells (EMD Millipore), and 5 ml Luria broth (LB) cultures were grown at 37°C for 16 h with appropriate antibiotics. Overnight cultures were used to inoculate 1-liter cultures in 2XYT media and grown at 37°C with

shaking at 180 RPM until $OD_{600}$ = 0.5. Protein expression was induced with the addition of 0.2 mM isopropyl $\beta$-d-1-thiogalactopyranoside (IPTG), the temperature was shifted to 20°C, and cultures were grown an additional 16 h. Cells were harvested by centrifugation and resuspended in lysis buffer (20 mM Hepes-NaOH, pH 7.0, 300 mM NaCl, 10% glycerol, 5 mM imidazole, 10 $\mu$M $ZnCl_2$).

Resuspended cells were lysed by sonication (Branson Sonifier), and then, the cell lysate was clarified by centrifugation. The supernatant was loaded onto an $Ni^{2+}$ affinity column (QIAGEN Ni-NTA Superflow) in lysis buffer, washed with wash buffer (20 mM Hepes-NaOH, pH 7, 25 mM imidazole, 300 mM NaCl, 5 mM $MgCl_2$, 10 $\mu$M $ZnCl_2$, 10% glycerol, and 5 mM $\beta$-mercaptoethanol), then eluted in elution buffer (20 mM Hepes-NaOH, pH 7, 500 mM imidazole, 300 mM NaCl, 5 mM $MgCl_2$, 10 $\mu$M $ZnCl_2$, 10% glycerol, 5 mM $\beta$-mercaptoethanol). Fractions were pooled and diluted to 100 mM NaCl through the addition of dilution buffer (20 mM Hepes-NaOH, pH 7, 25 mM imidazole, 5 mM $MgCl_2$, 10 $\mu$M $ZnCl_2$, 10% glycerol, 5 mM $\beta$-mercaptoethanol) before loading onto a cation-exchange column (HiTrap SP, #17-1152-01; Cytiva). Protein was eluted with a linear gradient from 100 mM to 1 M NaCl. For crystallography, His$_6$-tagged protein was incubated with TEV protease (expressed and purified in-house from vector pRK793, #8827; Addgene) (Kapust et al, 2001) for 48 h at 4°C. For biochemical assays, His$_6$-GST–tagged protein was not cleaved with TEV protease. TEV cleavage reactions were passed over a $Ni^{2+}$ affinity column a second time to remove His$_6$-tagged TEV protease and uncleaved Hop1, and the flow-through was collected and concentrated. Finally, proteins were passed over a size-exclusion column (Superdex 200 Increase 10/300 GL; Cytiva). Fractions were pooled and concentrated, then stored at –80°C (for biochemical assays), or buffer-exchanged into crystallography buffer (20 mM Hepes-NaOH, pH 7, 200 mM NaCl, 5 mM $MgCl_2$, and 1 mM tris(2-carboxyethyl)phosphine [TCEP]).

### X-ray crystallography

For crystallization of *S. mansoni* Hop1 CBR, purified protein at 10 mg/ml was mixed 1:1 with well solution containing 0.1 M MES, pH 6.0, 0.1 M sodium acetate, and 25% PEG 3350 in hanging drop format at 25°C. For crystallization of *P. miniata* Hop1 CBR, protein was subjected to reductive lysine methylation (Kim et al, 2008), and then, protein at 10 mg/ml was mixed 1:1 with well solution containing 0.1 M Bis-Tris, pH 6.5, 0.1 M potassium thiocyanate, and 17% PEG 3350 in hanging drop format at 25°C. In both cases, crystals were cryoprotected with an additional 30% xylitol and flash-frozen in liquid nitrogen. Diffraction data were collected at Advanced Photon Source beamline 24ID-C and processed with the RAPD pipeline, which uses XDS (Kabsch, 2010) for indexing and integration, AIMLESS (Evans & Murshudov, 2013) for scaling, and TRUNCATE for conversion to structure factors (Agirre et al, 2023). Anomalous sites representing protein-bound zinc ions were identified with hkl2map/SHELX (Sheldrick, 2010) and input into the Phenix Autosol pipeline (Liebschner et al, 2019) for phasing and automatic model building. Initial models were manually rebuilt in COOT (Emsley et al, 2010) and refined in phenix.refine (Adams et al, 2010). *S. mansoni* Hop1 CBR (1.45 Å resolution) was refined using positional and individual anisotropic B-factor refinement. *P. miniata* Hop1 CBR (1.84 Å resolution) was refined using positional,

individual isotropic B-factor, and TLS refinement (one TLS group per protein chain) (Table 1). Both models were refined with riding hydrogen atoms. All structural figures were created using PyMOL (version 3; Schrödinger, LLC). Surface charge representations were calculated with APBS (Jurrus et al, 2018).

### AlphaFold structure prediction

Structure predictions for Hop1 CBR–histone tail complexes were generated using the publicly available AlphaFold 3 server at https://alphafoldserver.com (Abramson et al, 2024). Template settings were set with the default PDB cutoff date of 29 September 2021, unless noted otherwise. ipTM scores for each model were extracted from the fold_job-name_summary_confidences_model-number.json files. NCBI/UniProt accession numbers for proteins used were as follows: XP_018653011 (*S. mansoni*), XP_038051536.1 (*P. miniata*), XP_001639673 (*N. vectensis*), XP_035662583 (*B. floridae*), XP_001639673 (*N. vectensis*), RDD46648 (*Trichoplax* sp. H2), XP_004363476 (*C. owczarzaki*), XP_009860628 (*C. intestinalis*), XP_002731279 (*S. kowalevskii*), XP_030838487 (*S. purpuratus*), A0A067R097 (UniProt; *Z. nevadensis*), A0A1X7V (UniProt; *A. queenslandica*), NP_012193 (*S. cerevisiae*), XP_001642921 (*Vanderwaltozyma polyspora*), XP_452539 (*K. lactis*), KAJ3363264 (*A. javanicus*), CAD25118 (*E. cuniculi*), KND01595 (*S. punctatus*), ELA45904 (*Vanderwaltozyma culicis*), KAI9179672.1 (*B. emersonii*), and EOB12410 (*N. bombycis*). Histone tail sequences used were SGRGKQGGKTRAKAKTRSSR (H2A), AKSAPAPKKGSKKAVTKTQK (H2B), ARTKQTARKSTGGKAPRKQL (H3), ART(K + N-trimethyllysine)QTARKSTGGKAPRKQL (H3K4me3), and SGRGKGGKGLGKGGAKRHRK (H4).

### Fluorescence polarization assays

To measure DNA binding, His$_6$-GST–tagged *S. mansoni* and *P. miniata* Hop1 CBR at the indicated concentrations were incubated with 50 nM fluorescein-labeled DNA duplex (5′-6-FAM-CTTATATCTGAATAGTCAGT-3′ annealed with 5′-ACTGACTATTCAGATATAAG-3′) in binding buffer (20 mM Tris–HCl, pH 8, 100 mM sodium glutamate, 5 mM $MgCl_2$, 3% glycerol, and 0.5 mM $\beta$-mercaptoethanol). Samples were incubated at 4°C for 30 min before reading fluorescence polarization on a Tecan Spark plate reader. Data were analyzed with GraphPad Prism version 10 (GraphPad software) using a one-site specific binding model.

### Reconstitution of NCPs

NCPs were reconstituted following the published protocols (Luger et al, 1999). Briefly, lyophilized *X. laevis* histones H2A, H2B, H3 (or H3K4me3), and H4 were purchased from the Histone Source at Colorado State University (https://histonesource-colostate.nbsstore.net/). The H3K4me3 histone contained K4C and C110A point mutants, and cysteine 4 was modified with a trimethyl-aminoethyl group according to published procedures (Simon et al, 2007) to mimic trimethyllysine. Histones were unfolded with incubation and shaking for 1 h at RT in unfolding buffer (6 M guanidine hydrochloride, 20 mM Tris–HCl, pH 7.5, 5 mM DTT). Histones were then added in equimolar ratio for 1 mg/ml final concentration. Histones were refolded into an octamer and dialyzed in refolding buffer (2 M NaCl, 10 mM Tris–HCl, pH 7.5, 1 mM

**Table 1.  Crystallographic data and refinement.**

| | S. mansoni Hop1 CBR | P. miniata Hop1 CBR |
|---|---|---|
| **Data collection** | | |
| Synchrotron/beamline | APS-24ID-C | APS-24ID-C |
| Date collected | 9/22/2021 | 9/22/2021 |
| Resolution (Å) | 64.96–1.45 | 82.95–1.84 |
| Wavelength (Å) | 1.2827 | 1.2827 |
| Space group | $P2_12_12_1$ | $P2_12_12_1$ |
| Unit cell dimensions (a, b, c) Å | 33.03, 64.96, 65.26 | 38.65, 106.96, 131.40 |
| Unit cell angles (α, β, γ)° | 90.00, 90.00, 90.00 | 90.00, 90.00, 90.00 |
| $I/\sigma$ (last shell) | 30.9 (5.3) | 14.7 (1.3) |
| $R_{merge}$ (last shell) | 0.08 (0.458) | 0.067 (0.827) |
| $R_{meas}$ (last shell) | 0.088 (0.573) | 0.080 (1.018) |
| $CC_{1/2}$ (last shell) | 0.997 (0.557) | 0.997 (0.454) |
| Completeness (last shell) % | 94.4 (50.3) | 98.9 (99.1) |
| Anomalous completeness (last shell) % | 90.3 (29.8) | 90.3 (75.4) |
| Number of reflections | 135,024 | 153,221 |
| Unique | 24,095 | 47,903 |
| Multiplicity (last shell) | 5.6 (2.3) | 3.2 (2.7) |
| **Refinement** | | |
| Resolution (Å) | 1.45 | 1.84 |
| No. of reflections | 24,041 (1,573) | 47,834 (4,585) |
| Working | 22,883 (1,475) | 45,414 (4,366) |
| Free | 1,158 (98) | 2,420 (219) |
| $R_{work}$ (last shell) (%) | 12.88 (20.83) | 19.79 (33.89) |
| $R_{free}$ (last shell) (%) | 15.83 (28.34) | 23.54 (35.61) |
| **Structure/stereochemistry** | | |
| No. of atoms | 1,400 | 7,970 |
| Solvent | 216 | 326 |
| Hydrogens | 1,186 | 3,780 |
| r.m.s.d. bond lengths (Å) | 0.014 | 0.0085 |
| r.m.s.d. bond angles (°) | 1.39 | 0.97 |
| SBGrid Data Bank ID | 1153 | 1156 |
| Protein Data Bank ID | 9MUG | 9MYP |

EDTA, and 5 mM $\beta$-mercaptoethanol). After refolding, histones were concentrated via centrifugation and loaded onto a size-exclusion column (200 16/600; Superdex) equilibrated in refolding buffer. Fractions were analyzed by SDS–PAGE, and fractions containing pure histones were pooled. For nucleosome reconstitution, the Widom 601 DNA sequence was amplified by PCR, purified, and concentrated. DNA was added to the histone octamer in 1:1.2 M ratio and dialyzed in 1.4 M KCl, 10 mM Tris–HCl, pH 7.5, 0.1 mM EDTA, 1 mM DTT for 1 h at 4°C. Low-salt buffer (10 mM KCl, 10 mM Tris–HCl, pH 7.5, 0.1 mM EDTA, 1 mM DTT) was slowly pumped into the high-salt buffer for a few hours and then replaced with low-salt buffer and dialyzed overnight. Nucleosomes were concentrated with a centrifugal filter and injected onto a size-exclusion column (Superose 6, 10/300 GL) in 20 mM Hepes, pH 7.5, 20 mM NaCl, 0.5 mM EDTA, 1 mM DTT. Fractions were run on (prerun at 0.5X TBE for 150 V for 1 h at 4°C) a 6% acrylamide TBE gel for 1 h at 150 V at 4°C. The gel was stained in SYBR Gold (1: 10,000) for 30 min while shaking in the dark. The gel was imaged with the ChemiDoc system, and pure nucleosomes were pooled and concentrated with a centrifugal filter.

### EMSAs

Recombinantly purified $His_6$-MBP-Hop1 CBR and reconstituted nonmethylated and methylated (H3K4me3) NCPs were incubated for 60 min at 4°C in binding buffer (20 mM Hepes, pH 7, 100 mM

NaCl, 100 $\mu$M ZnCl$_2$, 10% glycerol, 2 mM beta-mercaptoethanol). Samples were loaded onto 6% TB gel, pH 9.3 (gel was prerun at 20 mA for 1 h at 4°C), and run for 110 min at 20 mA at 4°C in running buffer (0.5X TB, pH 9.3). Gel was then incubated with shaking in SYBR Gold (S11494; Invitrogen) for 10 min in the dark. Gel was imaged on a ChemiDoc Imaging system (12003153; ChemiDoc).

## Data Availability

Raw diffraction data have been deposited at the SBGrid Data Bank (https://data.sbgrid.org) under accession codes 1153 (*S. mansoni* Hop1 CBR) and 1156 (*P. miniata* Hop1 CBR). Reduced data and final refined crystallographic models have been deposited at the RCSB Protein Data Bank (https://rcsb.org) under accession codes 9MUG (*S. mansoni* Hop1 CBR) and 9MYP (*P. miniata* Hop1 CBR).

## Supplementary Information

## Acknowledgements

The authors thank members of the Corbett laboratory for helpful discussions. The authors acknowledge support from the National Institutes of Health (K12 GM068524 to AA Rodriguez; R35 GM144121 to KD Corbett). This work is based upon research conducted at the Northeastern Collaborative Access Team beamlines, which are funded by the National Institute of General Medical Sciences from the National Institutes of Health (P30 GM124165). This research used resources of the Advanced Photon Source, a U.S. Department of Energy (DOE) Office of Science User Facility operated for the DOE Office of Science by Argonne National Laboratory under Contract No. DE-AC02-06CH11357.

### Author Contributions

AA Rodriguez: conceptualization, data curation, formal analysis, investigation, visualization, methodology, and writing—original draft, review, and editing.
AE Cirulli: investigation.
K Chau: investigation.
J Nguyen: investigation.
Q Ye: conceptualization, investigation, and methodology.
KD Corbett: conceptualization, formal analysis, funding acquisition, visualization, and writing—original draft, review, and editing.

### Conflict of Interest Statement

The authors declare that they have no conflict of interest.

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
