## [Reviewer comments · Life Science Alliance]

Life Science Alliance

Bipartite chromatin recognition by Hop1 from two diverged holozoa

Alyssa Rodriguez, Alessandro Cirulli, Katie Chau, Justin Nguyen, Qiaozhen Ye, and Kevin Corbett

DOI: <https://doi.org/10.26508/lsa.202503428>

Corresponding author(s): Kevin Corbett, University of California, San Diego

Review Timeline:

Submission Date:	2025-06-19
Editorial Decision:	2025-07-31
Revision Received:	2025-08-04
Accepted:	2025-08-07

Scientific Editor: Tim Fessenden

Transaction Report:

July 31, 2025

RE: Life Science Alliance Manuscript #LSA-2025-03428-T

Dr. Kevin D Corbett
University of California at San Diego (UCSD) and Ludwig Institute for Cancer Research
Cellular and Molecular Medicine
9500 Gilman Drive
CMM East Room 2058
La Jolla, CA 92093

Dear Dr. Corbett,

Thank you for submitting your manuscript entitled "Bipartite chromatin recognition by Hop1 from two diverged holozoa" to Life Science Alliance. This manuscript was evaluated by three expert reviewers, whose reports are attached. We appreciate your patience during the unusually long review process for your manuscript.

As you will see, all reviewers expressed enthusiasm for the intriguing new structural insights into Hop1-chromatin interactions, with implications for the mechanistic basis of meiotic recombination across diverged species. We invite you to implement improvements according to the minor suggestions made by Reviewers 1 and 3. We particularly encourage amending the text, as you see fit, in light of the final point made by Reviewer 3. We would be happy to publish your paper in Life Science Alliance pending these changes and final revisions necessary to meet our formatting guidelines.

- Please upload your main manuscript text as an editable doc file.
- Please upload your main and supplementary figures as single files.
- Please add a Running Title in our system.
- Please add a Summary Blurb/Alternate Abstract and a Category for your manuscript in our system.
- Please add an Author Contributions section to your main manuscript text and in our system.
- Please add a Conflict of Interest statement to your main manuscript text.
- Please add your main, supplementary figure, and table legends to the main manuscript text after the references section.
- It is recommended to exclude figures from the manuscript text and upload them separately.
- Please remove legends from the figures. They should be provided only in the manuscript file.
- Please add callouts for Figures S1A-D; S2B-I and S3 to your main manuscript text.

A. FINAL FILES:

-- Summary blurb (enter in submission system): A short text summarizing in a single sentence the study (max. 200 characters including spaces). This text is used in conjunction with the titles of papers, hence should be informative and complementary to

the title. It should describe the context and significance of the findings for a general readership; it should be written in the present tense and refer to the work in the third person. Author names should not be mentioned.

B. MANUSCRIPT ORGANIZATION AND FORMATTING:

Sincerely,

Reviewer #1 (Comments to the Authors (Required)):

Meiotic HORMA domain containing proteins are structural components of the chromosome axis that orchestrate multiple events during meiosis including the programmed induction of DNA double-strand breaks by Spo11 and their repair via inter-homolog recombination. The Corbett laboratory recently showed that the *S. cerevisiae* meiotic HORMAD protein, Hop1, contains a central chromatin-binding region, and that this domain is found across eucaryotes, albeit with distinct architectures. Here, Corbett and colleagues determined the crystal structure of two diverged holozoa (*S. mansoni* and *P. miniata*), structurally similar to each other, but distinct from that of *S. cerevisiae* Hop1 in that they lack a C-terminal HTH motif. They show that these CBR domains bind DNA in a canonical manner, different from that of yeast Hop1. In addition, they provide evidence that the PHD domains of *S. mansoni* and *P. miniata*, but not that of *S. cerevisiae*, may preferentially bind nucleosomes marked by H3K4me3, although this could unfortunately not be validated biochemically.

This is a very nice paper. The text is very concise, clear and straight to the point. The figures are beautifully designed, well labeled, and self-explanatory. The figure legends, materials and methods, and supplementary data contain all the relevant information. In my opinion, the paper can be published essentially in its current form. I have a few comments and suggestions, but none are particularly important.

- Figure 4: I'm wondering why the corresponding experiments were not shown for the *S. mansoni* CBR domain. It would also be better to show quantifications of these data.

- Potentially show a comparison of the DNA binding modes between *S. mansoni* and *P. miniata* CBR models and that of Hop1 structure in the supplement.

- Figure 2E,F: Please define error bars.

- P7 typo: Widom

- Finally, the authors made the choice to keep their discussion as brief as possible, and I respect that. Personally, I would have liked to read some more about the potential relevance of the H3K4me3 recognition for Hop1 function, why this may not apply to *cerevisiae*, and perhaps what to make of the lack of specificity for H3K4me3 in their biochemical assays. But this is up to them.

Reviewer #2 (Comments to the Authors (Required)):

In this manuscript, Rodriguez et al. determine the structure and chromatin-binding activities of Hop1 from two distantly related aquatic holozoans: *S. mansoni* and *P. miniata*. In budding yeast, Hop1 contains a central chromatin binding region (CBR) that binds bent nucleosomal DNA in a non-canonical manner through a composite interface formed by its PHD and wHTH domains. Interestingly, the domain organization and sequence of the CBR in budding yeast Hop1 have diverged significantly, including the addition of an HTH-C domain, raising the question of whether this reflects their distinct binding mechanism.

The authors now solve the crystal structures of the CBRs from *S. mansoni* and *P. miniata* and show that, unlike budding yeast Hop1, these CBRs bind DNA via the canonical binding interface typical of wHTH domain-containing proteins. Using AlphaFold modeling, they predict that the PHD domains of both CBRs likely bind to the N-terminal tail of histone H3. Finally, using EMSA, the authors demonstrate that *P. miniata* Hop1 CBR binds nucleosomes through both the PhD and wHTH domains.

The paper is concise and well written, with the structural and biochemical data that clearly support the main conclusion. I have no specific comments.

Reviewer #3 (Comments to the Authors (Required)):

In the present study the authors combine X-ray crystallography, AlphaFold3-based protein-protein interaction modelling and biochemical DNA and nucleosome binding assays to assess chromatin recognition by holozoan HORMAD domain-containing Hop1 proteins. This study is a follow-up to the author's 2014 article on Hop1 chromatin recognition in budding yeasts, now focusing on Hop1 of two more diverged holozoan species, a platyhelminthes and an echinodermata, both lacking a structural helix present in the yeast Hop1. The authors convincingly show that the structurally well-conserved Hop1 of both species bind dsDNA via their wHTH domain. Histone binding is only predicted by AlphaFold3 (AF3) modeling, but the contribution of the PHD domain in nucleosome binding is further confirmed biochemically, at least in one species' Hop1. Together, indeed indicating a bipartite chromosome recognition mechanism of holozoan Hop1 *in vitro*.

The study cleverly and convincingly combines crystallography with AF3 modeling and biochemical verification in a straightforward manner. I would support the acceptance of this manuscript with the following minor revisions:

- I was missing an information about why specifically these two species were chosen for the experiments and not, for example, more diverged holozoa.
- In Fig. 1A an information on the phylogenetic tree shown is missing. Please indicate that this is a cladogram, not an actual phylogeny.
- In Fig. 3F letters 'G' and 'H' are wrongly positioned and need to be moved up by one position.
- Although the authors showed a strong conservation in Hop1 structure between their two examined species that also indicates conservation of their modes of chromatin binding, I was wondering why the biochemical verification was only done for one species' Hop1. Showing conservation of bipartite chromatin binding biochemically in both species would increase significance of that finding here. Even if the experiment would not have worked, a short note would be beneficial. Here, the authors could also come back to their AF3-based predictions to see if the interaction with the histone tail would no longer be predicted by AF3 in their Hop1 mutants to kind of self-validate the power of combining protein-protein interaction prediction with biochemical verification.

Response to the Editor

In the section below, all editor queries are shown in black text, and our responses are shown in indented blue text.

Editor queries & instructions:

We invite you to implement improvements according to the minor suggestions made by Reviewers 1 and 3. We particularly encourage amending the text, as you see fit, in light of the final point made by Reviewer 3.

Please see the **Responses to Reviewers** below for detailed explanation of all changes made in response to suggestions from the reviewers.

Please upload your main manuscript text as an editable doc file.

Done

Please upload your main and supplementary figures as single files.

Done

Please add a Running Title in our system.

Done

Please add a Summary Blurb/Alternate Abstract and a Category for your manuscript in our system.

Done

Please add an Author Contributions section to your main manuscript text and in our system.

Done

Please add a Conflict of Interest statement to your main manuscript text.

Done

Please add your main, supplementary figure, and table legends to the main manuscript text after the references section.

Done

It is recommended to exclude figures from the manuscript text and upload them separately.

Done

Please remove legends from the figures. They should be provided only in the manuscript file.

Done

Please add callouts for Figures S1A-D; S2B-I and S3 to your main manuscript text.

Done. With the addition of a new **Figure S2** (based on a reviewer suggestions), the previous **Figures S2** and **S3** are now **Figures S3** and **S4**, respectively.

Manuscript Organization and Formatting

We have updated the manuscript with the section order as specified on the *Instructions to Authors* web page.

Final Files

We are supplying the final manuscript as an editable Word document (.docx), and all figures as PDF files.

Responses to Reviewers

We appreciate all three reviewers' careful analysis and critical questions. In the section below, all reviewer comments are shown in black text, and our responses are shown in indented blue text.

Reviewer #1:

Meiotic HORMA domain containing proteins are structural components of the chromosome axis that orchestrate multiple events during meiosis including the programmed induction of DNA double-strand breaks by Spo11 and their repair via inter-homolog recombination. The Corbett laboratory recently showed that the *S. cerevisiae* meiotic HORMAD protein, Hop1, contains a central chromatin-binding region, and that this domain is found across eucaryotes, albeit with distinct architectures. Here, Corbett and colleagues determined the crystal structure of two diverged holozoa (*S. mansoni* and *P. miniata*), structurally similar to each other, but distinct from that of *S. cerevisiae* Hop1 in that they lack a C-terminal HTH motif. They show that these CBR domains bind DNA in a canonical manner, different from that of yeast Hop1. In addition, they provide evidence that the PHD domains of *S. mansoni* and *P. miniata*, but not that of *S. cerevisiae*, may preferentially bind nucleosomes marked by H3K4me3, although this could unfortunately not be validated biochemically.

This is a very nice paper. The text is very concise, clear and straight to the point. The figures are beautifully designed, well labeled, and self-explanatory. The figure legends, materials and methods, and supplementary data contain all the relevant information. In my opinion, the paper can be published essentially in its current form. I have a few comments and suggestions, but none are particularly important.

- Figure 4: I'm wondering why the corresponding experiments were not shown for the *S. mansoni* CBR domain. It would also be better to show quantifications of these data.

Response

- Potentially show a comparison of the DNA binding modes between *S. mansoni* and *P. miniata* CBR models and that of Hop1 structure in the supplement.

This is a great idea. We have added this analysis as a new Supplemental Figure (Figure S2).

- Figure 2E,F: Please define error bars.

We have added the text "Error bars represent standard deviation from triplicate measurements" to the legend of Figure 2E and 2F.

- P7 typo: Widom

We have changed the typo "Wldom" to "Widom".

- Finally, the authors made the choice to keep their discussion as brief as possible, and I respect that. Personally, I would have liked to read some more about the potential relevance of the H3K4me3 recognition for Hop1 function, why this may not apply to *S. cerevisiae*, and perhaps what to make of the lack of specificity for H3K4me3 in their biochemical assays. But this is up to them.

We appreciate the reviewer's comment; however we chose not to speculate further given our current lack of direct data showing a clear preference for H3K4me3 chromatin binding in these proteins. This is certainly an area for future work.

Reviewer #2:

In this manuscript, Rodriguez et al. determine the structure and chromatin-binding activities of Hop1 from two distantly related aquatic holozoans: *S. mansoni* and *P. miniata*. In budding yeast, Hop1 contains a central chromatin binding region (CBR) that binds bent nucleosomal DNA in a non-canonical manner through a composite interface formed by its PHD and wHTH domains. Interestingly, the domain organization and sequence of the CBR in budding yeast Hop1 have diverged significantly, including the addition of an HTH-C domain, raising the question of whether this reflects their distinct binding mechanism.

The authors now solve the crystal structures of the CBRs from *S. mansoni* and *P. miniata* and show that, unlike budding yeast Hop1, these CBRs bind DNA via the canonical binding interface typical of wHTH domain-containing proteins. Using AlphaFold modeling, they predict that the PHD domains of both CBRs likely bind to the N-terminal tail of histone H3. Finally, using EMSA, the authors demonstrate that *P. miniata* Hop1 CBR binds nucleosomes through both the PHD and wHTH domains.

The paper is concise and well written, with the structural and biochemical data that clearly support the main conclusion. I have no specific comments.

We thank the reviewer for their support.

Reviewer #3:

In the present study the authors combine X-ray crystallography, AlphaFold3-based protein-protein interaction modelling and biochemical DNA and nucleosome binding assays to assess chromatin recognition by holozoan HORMAD domain-containing Hop1 proteins. This study is a follow-up to the author's 2014 article on Hop1 chromatin recognition in budding yeasts, now focusing on Hop1 of two more diverged holozoan species, a platyhelminthes and an echinodermata, both lacking a structural helix present in the yeast Hop1. The authors convincingly show that the structurally well-conserved Hop1 of both species bind dsDNA via their wHTH domain. Histone binding is only predicted by AlphaFold3 (AF3) modeling, but the contribution of the PHD domain in nucleosome binding is further confirmed biochemically, at least in one species' Hop1. Together, indeed indicating a bipartite chromosome recognition mechanism of holozoan Hop1 *in vitro*.

The study cleverly and convincingly combines crystallography with AF3 modeling and biochemical verification in a straightforward manner. I would support the acceptance of this manuscript with the following minor revisions:

- I was missing an information about why specifically these two species were chosen for the experiments and not, for example, more diverged holozoa.

We originally chose six Hop1 CBRs with PHD-wHTH architecture (from holozoa and holomycota) for biochemical/structural analysis, from *Podila verticillata* (fungi), *Rozella allomycis* (fungi), *Actinia tenebrosa* (Waratah anemone), *Schistosoma mansoni* (blood fluke), and *Patiria miniata* (sea star). *S. mansoni* and *P. miniata* Hop1 CBRs were simply the most amenable to *in vitro* analysis from this initial set. We hope to characterize other more diverged Hop1 CBRs, including from more diverged holozoa and from fungi, in future work.

- In Fig. 1A an information on the phylogenetic tree shown is missing. Please indicate that this is a cladogram, not an actual phylogeny.

We apologize for the confusion. This is not a cladogram; it is a subset of the phylogenetic tree shown in our prior work (Milano, Ur, et al. *EMBO J.* 2024). We have added an explanation of this point in the figure legend.

- In Fig. 3F letters 'G' and 'H' are wrongly positioned and need to be moved up by one position.

Thanks for catching this error; it is now fixed.

- Although the authors showed a strong conservation in Hop1 structure between their two examined species that also indicates conservation of their modes of chromatin binding, I was wondering why the biochemical verification was only done for one species' Hop1. Showing conservation of bipartite chromatin binding biochemically in both species would increase significance of that finding here. Even if the experiment would not have worked, a short note would be beneficial. Here, the authors could also come back to their AF3-based predictions to see if the interaction with the histone tail would no longer be predicted by AF3 in their Hop1 mutants to kind of self-validate the power of combining protein-protein interaction prediction with biochemical verification.

The reviewer is quite right that the *S. mansoni* and *P. miniata* Hop1 CBRs should show equivalent behavior in the nucleosome binding assays. We did perform nucleosome binding assays in parallel with both Hop1 CBRs. For technical reasons having to do with protein stability *in vitro* (especially for point mutants), the EMSA assays simply worked better and were more interpretable with the *P. miniata* Hop1 CBR. While results were broadly similar with the *S. mansoni* protein, we were unable to achieve sufficiently clear banding patterns with this protein, especially the point mutants thereof. We have added a sentence to the revised manuscript noting that the behavior of the *S. mansoni* protein limited our ability to perform this analysis.

The reviewer's question about consistency of *in vitro* assays with AlphaFold predictions is also interesting. To address this, we re-ran AlphaFold 3 predictions with the *P. miniata* Hop1 CBR (wild type and W376A mutant) and the histone H3 tail with H3K4me3 modification. The resulting structures and confidence values (AlphaFold ipTM) were essentially identical in both cases. The fact that AlphaFold could not readily predict a loss of H3k4me3 binding upon mutating W376 is not necessarily surprising, since AlphaFold was not designed to predict the effects of point mutations. Aspects of its design (notably, its reliance on multiple sequence alignments) mean that structure predictions are generally not strongly affected by point mutations. A more accurate prediction of the effects of a W376A mutation could come from a physics-based molecular dynamics simulation, but this approach is complex, also has downsides for interpretability, and is outside the scope of the current work. Because of the ambiguity of AlphaFold's predictions of the effects of point mutations, we chose not to add this analysis to the manuscript.

August 7, 2025

RE: Life Science Alliance Manuscript #LSA-2025-03428-TR

Dr. Kevin D Corbett
University of California, San Diego
Cellular and Molecular Medicine
9500 Gilman Drive
CMM East Room 2058
La Jolla, CA 92093

Dear Dr. Corbett,

Thank you for submitting your Research Article entitled "Bipartite chromatin recognition by Hop1 from two diverged holozoa". We appreciate your responses to the minor reviewer requests resulting in strengthened observations and discussion in this work. It is a pleasure to let you know that your manuscript is now accepted for publication in Life Science Alliance. Congratulations on this interesting contribution!

DISTRIBUTION OF MATERIALS:

Again, congratulations on a very nice paper. I hope you found the review process to be constructive and are pleased with how the manuscript was handled editorially. We look forward to future exciting submissions from your lab.

Sincerely,
